# Bacterial Lipopeptides Are Effective against Pear Fire Blight

**DOI:** 10.3390/microorganisms12050896

**Published:** 2024-04-29

**Authors:** Ihsan ud Din, Lina Hu, Yuan Jiang, Jie Wei, Muhammad Afzal, Li Sun

**Affiliations:** 1College of Life Sciences, Shihezi University, Shihezi 832003, China; ihsanuddin@stu.shzu.edu.cn (I.u.D.); hulina@stu.shzu.edu.cn (L.H.); 2Agricultural Scientific Institute of 2nd Division of Xinjiang Production and Construction Corps, Tiemenguan 841005, China; jorynks@163.com (Y.J.); 627weijie@sina.com (J.W.); 3College of Agriculture, South China Agricultural University, Guangzhou 510642, China; mafzal.kust@gmail.com

**Keywords:** biocontrol, *Priestia megaterium*, *Bacillus subtilis*, lipopeptides, TLC

## Abstract

Fire blight, a devastating disease caused by *Erwinia amylovora*, poses a significant threat to pear and apple trees in Xinjiang province, China. In an effort to combat this pathogen, we isolated 10 bacteria from various components of apple and crabapple trees and conducted screenings to assess their ability to inhibit *E. amylovora* in vitro. Through biochemical tests and partial 16S *rRNA* gene sequencing, we identified two promising strains, *Priestia megaterium* strain *H1* and *Bacillus subtilis* strain I2. These strains were then evaluated for their efficacy in biocontrol under controlled laboratory conditions, focusing on immature fruits and leaves. Remarkably, all selected antagonists exhibited the capability to reduce the severity of the disease on both fruit and leaves. *P. megaterium* strain H1 and *B. subtilis* strain I2 exhibited significant reductions in disease incidence on both immature fruits and leaves compared to the control. Specifically, on immature fruits, they achieved reductions of 53.39% and 44.76%, respectively, while on leaves, they demonstrated reductions of 59.55% and 55.53%, respectively. Furthermore, during the study, we detected the presence of lipopeptides, including *surfactin*, *iturins*, *bacillomycin* D, and *fengycins*, in the methanol extract obtained from these two antagonistic bacteria using thin-layer chromatography (TLC). Based on the results obtained, *B. subtilis* strain I2 and *P. megaterium* strain H1 exhibit considerable potential for controlling fire blight. However, further evaluation of their efficacy under natural field conditions is essential to validate their practicality as a biocontrol method.

## 1. Introduction

The bacterial disease fire blight is caused by the Gram-negative bacterium *Erwinia amylovora* [1]. Members of the subfamily *Maloideae*, particularly rosaceous plants like apple (*Malus* spp.) and pear (*Pyros* spp.) are severely afflicted by this disease. In warm and moist conditions, *E. amylovora* infects the plant hosts through the nectarthodes of the blooms or wounds in the leaves and twigs [2]. Once established in a plant, the bacteria can spread within the plant’s vascular system. As these bacteria accumulate in the xylem, the affected plant parts suffer from blight and ultimately die due to the blockage of water flow. Additionally, the presence of bacterial ooze, which consists of bacteria, polysaccharides, and plant sap, is a distinct characteristic of fire blight and is produced at the sites of infection. The pathogenic bacterium *E. amylovora* can infect various parts of the plant, including leaves, shoots, rootstocks, and fruits [3].

Insect pollinators and rainfall are the main routes of fire blight transmission [4]. Enterobacterium is a plant pathogen that induces a necrogenic disease. Initially, it manifests as black and necrotic lesions, eventually leading to plant mortality [5]. *E. amylovora*, the causative agent of this bacterial disease that was initially identified in the United States in the 1870s, has spread to Europe, Asia, Africa, China, and New Zealand [6].

Almost all cultivars of commercial pears and apples are vulnerable to *E*. *amylovora* [7]. *Streptomycin* and *oxytetracycline* can be used to effectively manage fire blight; however, this method has a risk of pathogenic strains becoming resistant to them [8]. To manage the disease, copper compounds are administered during the flowering stage. Nonetheless, the use of copper proves detrimental to the leaves, exhibiting phytotoxicity, and also leads to fruit irritation [9]. Biological control is particularly an effective strategy due to the significant risk of phytotoxicity and the causal agent’s tolerance to chemicals and antibiotics [10].

Bacteria such as *Bacillus subtilis*, *B. amyloliquefaciens* [11,12]. *Lactobacillus plantarum* [13], *Enterobacter* sp., and *Serratia* sp. [14] have demonstrated antagonistic activity against *E. amylovora*, indicating their potential as agents for controlling fire blight. Various mechanisms of action, including the production of secondary metabolites and competition for nutrients, have been suggested to explain their inhibitory effects on *E. amylovora* [15,16]. The use of antibiotics in agriculture, and antibiotic-resistant strains in numerous apple and pear growing regions, has evolved As a result, biocontrol agents (BCAs) for the prevention of fire blight have been studied for a long time [17]. BlightBanTM A506 [*Pseudomonas fluorescens* A506, isolated from leaves of pear trees] and *BlightBanTM* C9-1 [*Pantoea vagans* C9-1, isolated from apple stem tissue] are examples of commercial products that are currently available in the market [18].

Antipathogenic bacteria serve to lessen the severity and spread of disease by preventing the pathogen’s growth or competing for resources [19,20]. The antagonist bacteria’s inhibitory effects on *E. amylovora* are caused in a number of ways. These strategies include making antibiotics, vying for resources and space, causing systemic resistance in the host plant, and secreting enzymes that break down the pathogen’s cell walls [19]. For instance, research has revealed that *Pantoea agglomerans* (formerly known as *Erwinia herbicola*) produces *Pantocin* A, an antibiotic molecule that prevents the growth of *E. amylovora* [21]. Furthermore, it has been discovered that *Pseudomonas fluorescens* competes with *E. amylovora* for resources and space, restricting the pathogen’s growth [22]. *Bacillus subtilis* has been shown to induce systemic resistance in apple and pear trees, enhancing their defense against fire blight [23].

The potential of these antagonistic bacteria as biocontrol agents in orchard environments has been thoroughly investigated. They can be used as a component of integrated pest management plans to lessen the need for chemical pesticides and their negative environmental effects. Growers can successfully control fire blight and save their fruit crops by utilizing the inhibitory effects of antagonist bacteria [24].

The objective of this study is to explore the potential of endophytic bacteria, specifically *P. megaterium* and *B. subtilis*, isolated from apple and crabapple leaves, as antagonists against *E. amylovora*, a pathogen causing fire blight in these fruits. The study aims to assess the efficacy of these bacterial isolates both in vitro and in vivo as biological control agents (BCAs) against *E. amylovora*. This research aims to contribute to the sustainable and environmentally friendly approach of using natural antagonists, such as bacteria, particularly *Bacillus*, to control pests, pathogens, and diseases in plants. By investigating mechanisms like antibiosis, competition, and induced systemic resistance, the study sheds light on the distinct characteristics and behaviors of these bacteria, offering valuable insights into their potential as effective antagonists in various biological contexts.

## 2. Materials and Methods

### 2.1. Identification and Isolation of the Bacterium That Causes Fire Blight (E. amylovora C1)

We used as standard strains of *E. amylovora* C1 isolated by our research group [25]. To nurture and propagate *E. amylovora* cultures, we employed nutrient agar (NA), which consists of beef extract, peptone, agar, and sodium chloride, with a pH range maintained between 7.2 and 7.4 [26].

#### Preparation of a Fire Blight Pathogenic Bacterial Suspension

The method outlined by [27] was employed with minor adjustments to revive the *E. amylovora* C1 strain from the −80 °C freezer, allowing it to reach room temperature. Following this, a small aliquot of bacterial solution was transferred using an inoculation ring onto NA plates and then incubated for 24 h at 28 °C. Subsequently, robust colonies were selected and sub-cultured into a nutrient broth (NB), which was placed on a constant shaker at 28 °C and agitated at 210 rpm for an additional 24 h. Throughout the incubation period, growth was continually monitored to achieve an optical density OD600nm = 1 (approximately 1.0 × 10^8^ CFU/mL).

### 2.2. Isolation and Identification of Leaves Endophytic Antagonist Bacteria

The methodology described by [28] with slight modification was used to collect leaf samples from healthy apple and crabapple trees at various sites within Shihezi University, Xinjiang Province, China, during May to September of 2022 and 2023. The objective was to procure potential bacterial antagonists for further investigation. Upon collection, leaf samples were either immediately processed or temporarily refrigerated in plastic bags to maintain freshness. Leaf tissue sections were macerated in sterile-distilled water (2–3 milliliters) for 30 min to create suspensions for subsequent analysis. These suspensions were streaked onto NA plates, chosen for their ability to support bacterial growth. Streaking was crucial for isolating potential bacterial antagonists. The NA plates were then incubated at temperatures ranging from 26 to 28 °C for 48–72 h to allow observation and cultivation of bacterial colonies present in the leaf samples, facilitating their subsequent identification and study.

### 2.3. Antagonist and Pathogenic bacteria (E. amylovora) Screening

The methodology outlined by [14] with slight adjustments involves cultivating bacterial cultures obtained from *E. amylovora* C1 and leaf samples in a nutrient-rich NB liquid medium (3.0 g/L beef extract, 5.0 g/L peptone, 5.0 g/L NaCl, pH 7.2–7.4) at 28 °C with gentle agitation at 210 rpm. Following a 24 h incubation period, the bacterial suspension was diluted to achieve an optical density OD600nm = 1 (approximately 1.0 × 10^8^–1.0 × 10^9^ CFU/mL).

To assess antimicrobial activity, 300 µL of pathogenic bacterial suspension (approximately 2 × 10^8^ CFU/mL) was spread onto nutrient agar plates and allowed to stand for 5 min. Subsequently, paper discs saturated with antagonistic bacteria suspensions adjusted to around 10^8^ CFU/mL were placed on the plates. The plates were then incubated at 27 °C for 48–72 h, during which inhibition zones developed around the paper discs due to antimicrobial action. These zones were measured to evaluate the inhibition of bacterial growth [10]. Sterile water served as a control on pathogen-inoculated plates to confirm that any inhibitory effects observed were solely attributable to the antagonistic bacteria.

### 2.4. Physiological and Biochemical Characteristics of Antagonistic Bacteria

Antagonist bacteria were subjected to a series of tests to evaluate their physiological and biochemical reactions. These tests encompassed a range of important reactions, including Gram staining, starch hydrolysis, M.R (Methyl Red) test, glucose fermentation, sucrose fermentation, motility test, V.P (Voges–Proskauer) test, gelatin test, and citrate test, as outlined in [29].

### 2.5. Identification of Antagonist Bacterial Strains via Phylogenetic Examination of Their 16S rDNA Sequences

The bacterial strains were subjected to molecular identification through sequencing of the 16S rRNA gene segment. Genomic DNA extraction was performed using the Genomic DNA Kit from Urumqi Youkang Biotechnology Co., Ltd., Urumqi, China. Amplification of the 16S rRNA gene utilized the primer pair 27F (5′-AGAGTTTGATCCTGGCTCAG-3′) and 1492R (5′-GGTTACCTTGTTACGACTT-3′) [30]. PCR reactions were performed in a 20 μL volume according to the protocols outlined in [31], with slight modifications, specifying components and cycling conditions. Following successful amplification, DNA fragments were purified and sequenced by Urumqi Youkang Biotechnology Co., Ltd. The sequence data were uploaded to the NCBI database with reference numbers (PP453776.1 and OR511440.1) for comparative analysis with reference sequences. Phylogenetic analysis was conducted using the neighbor-joining (NJ) method with 1000 bootstrap iterations, implemented in MEGA 7.0 software. Sequence similarity calculations and multiple alignments were performed using Clustal W Mega 7.0. The phylogenetic relationships between *Priestia* and *Bacillus* strains were illustrated using neighbor-joining and MEGA 7.0, with 16S rDNA sequences as the dataset, supported by 1000 rounds of bootstrap resampling.

### 2.6. Identifying the Specific Genes Responsible for Producing Known Antimicrobial Substances

The screening process involved the utilization of specific primers for targeted genes, with PCR programs tailored to the individual annealing temperatures. The subsequent gel electrophoresis allowed for the observation and comparison of amplified DNA fragments, providing insights into the presence of gene targets related to antimicrobial compounds in the studied bacteria.

To detect the presence of gene targets associated with antimicrobial compounds in both *B. subtilis* and *P. megaterium* antagonist bacteria, PCR screening was conducted using specific primers outlined in (Appendix A) [32,33]. The genes *FenD*, *ItUc*, *YndJ*, *SrfAA*, *Spas*, and *QK1* were specifically targeted. The PCR program employed for this analysis featured an annealing temperature of 52 °C for *fenD*, *ituC*, and *yndJ*, while *srfAA*, *Spas*, and *QK1* utilized temperatures of 62 °C, 58 °C, and 55 °C, respectively. Following the PCR reactions, the amplified products were separated on a 1.8% agarose gel in 0.5× Tris-borate EDTA (TBE) buffer for 2.5 h at 140 V. Visualization of the DNA bands was achieved by staining the gel with ethidium bromide. Size comparisons were conducted using a 5000 DNA marker provided by Nanjing Vazyme Biotech Company (Nanjing, China). Gel images were captured using an imaging system (Bio-Rad GelDoc EZ Gel Imaging System Thermo Fisher Scientific (Shanghai), Ltd., Shanghai, China).

### 2.7. Antibacterial Activity Observation

To initiate the experiment, a pure strain of the pear fire blight bacterium *E. amylovora* C1 was cultured, ensuring the logarithmic growth phase was reached. The bacterial concentration was adjusted to the desired level by measuring the optical density and diluting the suspension accordingly to OD600nm = 1 (approximately 1.0 × 10^8^ CFU/mL). Nutrient agar medium was prepared by mixing beef extract, peptone, and NaCl, adjusting the pH to 7.2–7.4, and sterilizing via autoclaving. Agar plates were inoculated with 100 μL of bacterial suspension, ensuring an even distribution. Three holes were created in each plate using a sterile hole puncher, then 200 μL of antibacterial filtrate was added into the holes for treatment groups and sterile water for the control. Plates were incubated at 28 °C for 24 h for bacterial growth and inhibition zone development. The inhibition zone diameters were measured meticulously following established techniques referenced in [34]. Statistical analysis was performed to compare results between the antibacterial filtrate and the control. Conclusions were drawn from outcomes, highlighting implications for pear fire blight bacterium susceptibility to the antibacterial agent. Consistency was maintained through three replicates for each treatment to ensure robust data.

### 2.8. Assessing the Antibacterial Efficacy on Korla Fragrant Pear Leaves and Fruits

Potentially therapeutic and preventive activities of *B. subtillis* strain I2 and *P. megaterium* strain H1 against *E. amylovora* on detached pear tissues, such as leaves and fruits of the “Korla fragrant” pear, were investigated in the laboratory setting. To evaluate the effectiveness of the treatment, we used *streptomycin*. In order to evaluate the antibacterial properties of strains H1 and I2, we first used tap water to wash the removed leaves of the “Korla fragrant” pear. After being submerged in 3% bleach for 10 min, they were cleaned three times using sterile disinfected water (SDW). Using a syringe, we injected 100 μL of SDW into the intercellular gaps on the hollow petiole of mature leaves as a negative control. To maintain consistency, SDW was sprayed on all of the leaves [10,35].

The positive control was obtained by injecting 100 μL suspensions of pathogenic bacteria *E. amylovora* C1 (1.0 × 10^8^ CFU/mL) in place of SDW, using the same technique. We sprayed the leaves with suspensions of antagonistic bacteria strains H1 and I2 (1.0 × 10^8^ CFU/mL) and streptomycin solution separately after 48 h of incubation. The leaves were sprayed separately with *streptomycin* or strains H1 and I2 for protective tests. We injected 100 μL of pathogenic bacterial suspensions into the leaves after a further 48 h. We treated the leaves differently between the curative trials and the protective ones. We created a treatment control in which suspensions of strains H1 and I2 were sprayed onto leaves in order to observe the impact of the strains on the leaves. The suspensions of strains H1 and I2 were sprayed on the leaves once more after a 48 h period. After that, the treated leaves were put in a climate chamber at a temperature of 28 °C and a light-to-dark ratio (12:12 L:D), and symptoms were measured starting three to fifteen days after the inoculation (dpi). Three leaves were used in each treatment, and the trials were repeated three times. By calculating the percentage of leaf area impacted, the severity of the disease was determined [36]. Similarly, experimentation on pear fruits involved surface sterilization, maintenance in a sterile environment, and inoculation with either sterile water or a pathogenic cell suspension. Protective and curative assays were conducted using bacterial suspensions. Disease progression was monitored after 15 days using an infection index scale. The effectiveness of treatments was determined using the Abbott formula [36].

The disease severity (DS) is expressed as a percentage using the formula
DS (%) = (a/b) × 100,
where “a” represents the length of the blighted portion of the plant organ in centimeters, and “b” represents the total length of the plant organ in centimeters.

To determine the effectiveness (E) of the applications employed in the experiments, the Abbott formula is utilized and can be stated as follows:
E (%) = ((K − U)/K) × 100,
where “E” denotes effectiveness, “K” represents the percentage disease severity of the control plant organ, and “U” represents the percentage disease severity of the treated plant organ.

### 2.9. Antibiotic Sensitivity Test

A methodology was adopted using an NA medium to assess the susceptibility of *P. megaterium* and *B. subtilis* to eight antibiotics at a concentration of 30 μg/mL. The antibiotics chosen for the study were *tetracycline*, *kanamycin*, *streptomycin*, *cefotaxime*, *erythromycin*, *ampicillin*, *penicillin*, and *ciprofloxacin*. The antibiotics underwent sterilization by the process of filtering. Afterwards, 5 μL of each antibiotic solution was placed onto Whatman paper discs. These discs, infused with antibiotics, were placed strategically on the NA medium. The experimental setup underwent 24 h incubation at 28 °C to simulate optimal bacterial growth and antibiotic interaction conditions. Following incubation, the agar plates were inspected for observable phenomena [5]. The presence of inhibition zones around the paper discs indicated bacterial susceptibility to antibiotics, while their absence indicated resistance. This methodological framework, grounded in microbiological principles, offers insights into the antibiotic susceptibility of *P. megaterium* and *B. subtilis* compared to standard references as demonstrated in (Appendix A) [37].

### 2.10. Growth Optimization of Isolates

We investigated various parameters affecting the growth dynamics of *B. subtilis* and *P. megaterium*. Initially, growth curves were constructed using a UV–Vis spectrophotometer to monitor the turbidity of nutrient broth cultures over time, elucidating temporal aspects of bacterial growth and interactions. Subsequently, we explored the temperature sensitivity of these bacteria across a range of temperatures (28 ± 2 °C to 36 ± 2 °C) in nutrient broth medium, identifying optimal growth conditions [38]. pH sensitivity was evaluated by culturing the bacteria in a nutrient broth medium across a pH range of 4 to 8, revealing their adaptability to varying pH levels. Furthermore, in vitro growth dynamics were examined by subjecting the bacterial cultures to different rpm values (150 to 210), indicating the impact of rotational speed on proliferation. Finally, we also assessed bacterial growth responses to varying inoculation amounts (1% to 4%) within the nutrient broth medium, providing insights into adaptive responses to different concentrations and environmental conditions [39,40]. These analyses contribute to a better understanding of the growth characteristics and environmental adaptability of *B. subtilis* and *P. megaterium*.

### 2.11. Thin-Layer Chromatography (TLC)

The thin-layer chromatography (TLC) technique, as outlined by [11], was employed to identify the antibacterial constituents within the extracted supernatant. Separation was achieved using a mixture of chloroform, methanol, and water (65:25:4, *v*/*v*/*v*), with visualization of spots accomplished by spraying the TLC plate with ninhydrin solution. The characterization of individual molecules was conducted through the calculation of their retention factor (Rf) values, with each experiment being replicated three times. The following lipopeptides were used as standard compounds: *surfactin*, *iturins*, *bacillomycin* D and *fengycins*. Table 1 provides specifics on how their corresponding retention factors (RF) were determined using TLC analysis [41,42].

## 3. Results

### 3.1. Identification of Bacterial Isolates

#### 3.1.1. DNA-Based Identification

In the exploration of microbial diversity on apple and crabapple tree leaf surfaces, 10 bacterial specimens were isolated and screened for inhibitory potential against pathogens in vitro. Two strains displayed promising inhibitory activity. The 16S rRNA genes of these strains were partially sequenced, revealing a high similarity of 98–99% with known species in the GenBank database. (Figure 1A).

Analysis of the nearly complete 16S rRNA gene sequences indicated a resemblance to those of *P. megaterium* and *B. subtilis*. Phylogenetic trees were constructed using a neighbor-joining method, illustrating the genetic relationships between the isolated strains and closely related bacteria (Figure 1B).

#### 3.1.2. Characterization of Antagonistic Bacterial Species: Morphological, Physiological, and Biochemical Traits

The growth patterns of antagonist bacteria on the NA medium are depicted in Appendix A, while Table 2 provides a summary of the bacterium’s morphological, physiological, and biochemical features. The Gram staining technique demonstrates that both antagonistic bacteria have a Gram-positive nature and a rod-shaped morphology. This is evident from the white-coloured colonies observed (Appendix A). In addition, both bacterial isolates exhibited motility and tested positive for glucose, catalase, gelatin liquefaction, and starch hydrolysis. Both isolates exhibited a negative reaction in the MR (Methyl Red) test, while displaying a positive result in the V.P (Voges-Proskauer) test.

### 3.2. Antibiotic Sensitivity Test

The antibiotic resistance of both antagonistic bacteria was evaluated using the disk-diffusion method. As shown in (Figure 2A,B), the results show moderate sensitivity of *P. megaterium* H1 to *erythromycin* and moderate sensitivity of *B. subtilis* I2 to both *erythromycin* and *ciprofloxacin*. Additionally, both antagonistic bacteria exhibited susceptibility to *ampicillin*, *penicillin*, *cefotaxime*, *kanamycin*, *streptomycin*, and *tetracycline*.

### 3.3. Factors Affecting Antagonistic Bacterial Growth Sensitivity Analysis

Both antagonistic bacteria, *B. subtilis* strain I2 and *P. megaterium* H1, grow well between pH 5 and 7, with a maximum OD_600_ value of 1.561 at pH 5 for *B. subtilis* I2 and 2.029 for *P. megaterium* H1 (Figure 3A). When the inoculum amount is 3%, the maximum OD_600_ value is 2.031 for *P. megaterium* H1 and 2.112 for *B. subtilis* strain I2, which is the optimal inoculum amount (Figure 3C). Different temperatures have a greater impact on the growth of both antagonistic bacteria. The maximum OD_600_ value is 2.068 at 34 °C for *B. subtilis* I2 and 2.028 at 30 °C for *P. megaterium* H1, which is the optimal growth temperature for both (Figure 3B). There are certain differences in the OD_600_ values under different shaker speed conditions. In the range of 150–210 r/min, the OD_600_ value continues to increase with the increase of the rotation speed. When the rotation speed is 210 r/min, the maximum OD_600_ value is 2.165 for *B. subtilis* I2, but for *P. megaterium* H1, 150 r/min is determined to be the optimal rotation speed because its maximum OD_600_ value at this rotation speed is 2.028 (Figure 3D).

### 3.4. Growth Curve of Antagonist Bacteria

The growth curves of the antagonist bacteria *B. subtilis* strain I2 and *P. megaterium* H1, monitored over a total period of 18 h with measurements taken at 2 h intervals, exhibited typical bacterial growth dynamics, consisting of lag, exponential, and stationary phases. For the H1 strain, the lag phase persisted for the initial 4 h, followed by a rapid increase in growth during the exponential phase, which extended from 4 to 12 h. By the end of the exponential phase, the growth rate began to slow down, indicative of the approaching stationary phase, which continued until the end of the observation period. Similarly, the I2 strain demonstrated a lag phase lasting for the initial 6 h, followed by an exponential growth phase extending from 6 to 14 h. Subsequently, the growth rate slowed, transitioning into the stationary phase, which persisted until the end of the monitoring period. These observations illustrate the characteristic growth dynamics of bacterial populations, highlighting distinct phases in their growth patterns over time (Appendix A).

### 3.5. In Vitro Antagonistic Activity of B. subtilis I2 and P. megaterium H1 against E. amylovora

Ten bacterial strains were obtained from apple and crabapple plants. Out of these, two strains shown strong antagonistic activity against *E. amylovora* using the plate confrontation method (Figure 4A). The range of the inhibition zone diameters was between 15.95 ± 0.65 and 25.90 ± 0.25 mm. *B. subtilis* I2 exhibited significantly greater inhibitory activity against *E. amylovora* compared to *P. megaterium* H1. The inhibition zone diameter of *B. subtilis* I2 was measured at 25.90 ± 0.25 mm (Figure 4B), while *P. megaterium* H1 had a smaller diameter of 15.95 ± 0.65 mm. These two strains were chosen for additional analysis and identification.

### 3.6. Identifying the Specific Genes Responsible for Producing Known Antimicrobial Substances

PCR amplification was conducted on the DNA of antagonistic bacteria *B. subtilis* I2 and *P. megaterium* H1 targeting six antibiotic-related genes from *Bacillus* and *Prietia*. Positive PCR results were exclusively obtained for three antimicrobial genes common to both bacterial strains, as elaborated and depicted in (Figure 5A–C). Among the detected genes, *iturin* synthesis gene *ituC*, *fengycin* synthesis gene *fenD*, and *subtilin* synthesis gene *QK1* were identified. However, genes *srfAA*, *YndJ*, and *Spas* were not detected in this study. This suggests a distinct genetic makeup and antimicrobial potential between the strains, highlighting the variability in antibiotic gene expression among antagonistic bacteria.

### 3.7. Antibacterial Filtrate Activity

The antibacterial activity of *P. megaterium* H1 and *Bacillus subtilis* against *E. amylovora* was evaluated using the disk-diffusion method. Results show that the cell-free supernatant of both *P. megaterium* H1 and *Bacillus subtilis* displayed antimicrobial efficacy in comparison to bacterial cell disruption (Appendix A). This observation suggests that the antimicrobial activity can be attributed to the extracellular secondary metabolites secreted by *P. megaterium* H1 and *Bacillus subtilis* I2.

### 3.8. Efficacy of Bacterial Antagonists on Detached Pear Fruits and Leaves

The results showed significant reductions in disease symptoms and pathogen proliferation in fruit treated with strains of *B. subtilis* I2 and *P. megaterium* H1, comparable to the effectiveness of *streptomycin*. These strains exhibited a notable reduction in fruit infection, with *B. subtilis* I2, *P. megaterium* H1, and streptomycin reducing infection by 53.39%, 44.76%, and 49.69%, respectively (Figure 6A and Figure 7A,B).

Similarly, in detached pear leaf experiments, thorough disinfection preceded treatments with bacterial suspensions or streptomycin. Observations over a 15-day period revealed significant reductions in foliar disease symptoms and pathogen proliferation in leaves treated with strains *P. megaterium* H1 and *B. subtilis* I2. *B. subtilis* I2 and *P. megaterium* H1 demonstrated high efficacy comparable to *streptomycin*, with reductions in foliar infection by 59.55%, 55.53%, and 56.69%, respectively (Figure 6B and Figure 7C,D). These findings underscore the potential of *B. subtilis* I2 and *P. megaterium* H1 as effective biological control agents against fire blight, offering promising alternatives to chemical interventions.

Significantly reduced symptoms of disease and pathogen development were seen in pear tissues treated with *B. subtilis* I2 and *P. megaterium* H1 in all in vivo experiments, compared to the positive control (*p* < 0.05). Moreover, the efficacy of protective and curative treatments using *B. subtilis* I2 and *P. megaterium* H1 against *E. amylovora* were comparable to that of *streptomycin*, suggesting their potential as biological control agents for managing fire blight.

### 3.9. Analyzing Lipopeptides with TLC

Thin-layer chromatography (TLC) was employed for the separation and preliminary identification of lipopeptides within methanol extracts, utilizing *ninhydrin* solution spray for visualization. Lipopeptides such as *fengycins*, *iturins*, *bacillomycin* D, and *surfactins* were successfully separated and identified using this method.

To identify the compounds responsible for inhibiting the growth of *E. amylovora*, methanol extracts of lipopeptides derived from the antagonistic bacteria *B. subtilis* I2 and *P. megaterium* H1 were analyzed by TLC. The analysis revealed four active compounds from both strains, which exhibited inhibitory effects on *E. amylovora* growth. Two of these active compounds were found in both *B. subtilis* I2 and *P. megaterium* H1, displaying Rf values consistent with *iturin-* and *surfactin*-like lipopeptides (0.56 to 0.64). Additionally, *bacillomycin* D (Rf 0.35) was identified in *B. subtilis* I2, while *fengycins* (Rf 0.43) were detected in *P. megaterium* H1 (Figure 8A,B). The findings corroborate previous reports by [41,42].

## 4. Discussion

The fragrant pear industry in Xinjiang, China, faces significant challenges stemming from the proliferation of pests and diseases such as pear scab, rust, black spot, and fire blight. Controlling pear fire blight poses significant challenges due to the resilient nature and wide dispersion of *E. amylovora* as well as the diverse range of tissues it infects. Currently, there is no singularly effective chemical or control method for this disease [43]. Traditional approaches involve pruning affected branches and applying copper-based compounds or antibiotics like *streptomycin* and *oxytetracycline* [5,44]. However, these methods have limitations; pruning may be ineffective in severe cases, while excessive copper application can lead to plant toxicity and yield reduction. Prolonged antibiotic use also carries the risk of bacterial resistance and environmental contamination. As a result, there is growing interest in biological control methods as promising alternatives. These approaches leverage natural agents or mechanisms to suppress the growth and spread of *E. amylovora*. By harnessing biological control agents such as antagonistic microbes or bacteriophages, researchers aim to develop sustainable strategies that minimize environmental impact and reduce the risk of resistance development. This shift towards biological control reflects a recognition of the limitations and drawbacks associated with conventional chemical treatments, signaling a proactive effort to explore innovative solutions for managing pear fire blight effectively [10,45].

In recent years, challenges in the fragrant pear industry have been exacerbated by various factors, including extensive planting, dependence on a narrow range of cultivars, and frequent international fruit trade. The combined effects of these factors have significantly impeded the sustainability and productivity of the fragrant pear industry [46].

During the initial phase, in the orchards situated in Korla, Xinjiang, distinctive symptoms of dark brown wilting were noted on the tender shoots of pear trees. These symptoms bore a resemblance to those typically associated with pear fire blight. In efforts to pinpoint the causative agent responsible for this manifestation, bacterial samples were carefully isolated from the affected branches and subjected to thorough identification procedures.

In this study, we isolated 10 distinct bacteria from both apple and crabapple leaves, and identified two strains that exhibited antagonistic behavior against *E. amylovora*, the causative agent of fire blight. Using a combination of 16S rRNA analysis, biochemical assays, and microscopy, the antagonistic bacteria were classified as *Bacillus* and *Priestia* species. Notably, *B. subtilis* strain I2 displayed the most potent antagonistic activity against *E. amylovora*. Subsequent in vitro experiments evaluated the protective and therapeutic efficacy of *P. megaterium* H1 and *B. subtilis* strain I2 on pear tree tissue. The findings demonstrated that both strains effectively mitigated pear fire blight infections in leaves and fruits. Furthermore, the protective application of the antagonistic bacteria was more effective in preventing infection compared to therapeutic intervention. These results highlight the potential of these bacterial strains as biocontrol agents for managing fire blight in pear orchards.

While there is not much data on the ability of *P. megaterium* and *B. subtilis* to suppress fire blight in pears, new research has investigated their potential as biocontrol agents. *P. megaterium* and *B. subtilis* were injected onto the leaves and fruits of solitary pear trees in experiments to evaluate their efficacy in protective and therapeutic roles. According to the data, both bacterial species showed promise in lowering the frequency of fire blight, with protective treatment producing better results than therapeutic intervention. Notably, *B. subtilis* strain I2 has shown great promise as a biocontrol agent for controlling pear orchard fire blight. These results highlight the potential of *B. subtilis* and *P. megaterium* as effective fire blight control agents in pear orchards.

Studies has showed that *Bacillus* species are effective against plant infections because they produce antimicrobial chemicals [47]. *Bacillus* species are capable of synthesizing many bioactive compounds, including non-ribosomal peptides, lipopeptides, polyketides, siderophores, and bacteriocins [48]. *Bacillus* produces lipopeptides, which can be classified into three main families: *fengycins*, *surfactins*, and *iturins* [49]. *Iturins* have strong antifungal effects against various types of yeast and filamentous fungi, but their antibacterial effects are limited. Fengycins have effective antifungal activity, especially against filamentous fungi, and recent studies suggest they also have antibacterial properties [50]. Sur-factins are known for their significant bactericidal activity [51]. The disease control achieved with lipopeptides involves both direct interaction with the biological membranes of bacterial and fungal pathogens and indirect stimulation of systemic resistance in plants [52]. *P. megaterium* and *B. subtilis* strains generate a wide range of bioactive lipopeptides, such as *surfactins*, *lichenysin*, *iturin* A, and *fengycins* A and B [53,54].

In this study, both the cell-free supernatant and methanol extract derived from *P. megaterium* H1 and *B. Subtilis* I2 exhibited antibacterial properties, indicating the secretion of bioactive compounds, potentially secondary metabolites, with hydrophobic characteristics. TLC analysis revealed the presence of lipopeptides in the methanol extract, as evidenced by the formation of a blue-purple substance from *P. megaterium* H1, with corresponding RF values of 0.231, 0.430, 0.569, and 0.641, while for *B. subtilis* I2, they were 0.251, 0.271, 0.359, 0.541, and 0.630. These RF values imply that the methanol extract contains lipopeptides (*surfactins*, *iturins*, *bacillomycin* D, and *fengycins*), consistent with prior research [11,42,55].

PCR results were obtained exclusively for three antimicrobial genes common to both antagonistic bacteria, which produce the *subtilin*, *fengycin*, and *iturin* A found in the methanol extract. These lipopeptides possess antibacterial, antifungal, and antiviral activities, suggesting their potential application in the prevention and treatment of pear fire blight [32].

This study found that *P. megaterium* strain H1 and *B. subtilis* strain I2 effectively inhibited the growth of *E. amylovora* on detached pear leaves and fruits. However, additional research is required to examine the effectiveness of strain H1 and I2 in combating fire blight disease in natural environmental settings. In conclusion, the broad-spectrum antagonistic activity exhibited by *P. megaterium* H1 and *B. subtilis* I2 makes them potential bacteria for the development of novel biocontrol agents to combat fire blight disease.

## 5. Conclusions

The current study identified two out of ten distinct bacteria isolated from the leaves of apple and crabapple plants that exhibited antagonistic behavior toward *E. amylovora*, the pathogen responsible for fire blight in pears. These bacteria, namely *P. megaterium* H1 and *B. subtilis* I2, were found to release unique peptide antibiotics, such as *fengycin*, *subtilin*, and *iturin* A, into the environment to inhibit the activity of *E. amylovora* C1. Further analysis revealed the presence of several antibacterial substances, including *surfactin*, *iturins*, *bacillomycin* D, and *fengycins*, in the methanol extract obtained from these two antagonistic bacteria using TLC. This finding suggests the potential these bacteria and their antibiotic compounds have in controlling fire blight disease in pear orchards.

## Figures and Tables

**Figure 1 microorganisms-12-00896-f001:**
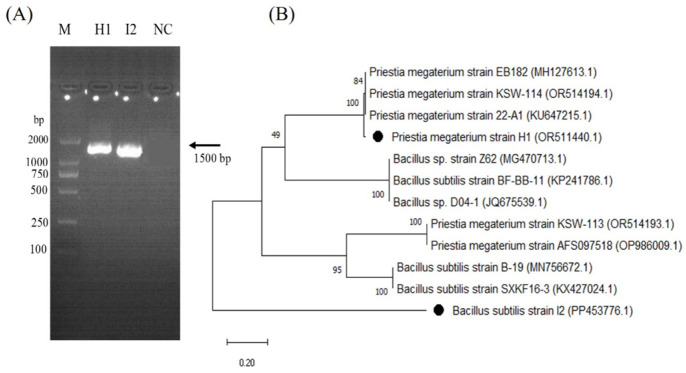
(**A**) I6S rRNA gene PCR detection of *P. megaterium* H1 and *B. subtilis* I2. (**B**) Phylogenetic tree of partial 16S rRNA gene sequences of antagonistic bacteria along with the sequences from selected references strains. The analysis was conducted with MEGA 11 using a neighbor-joining method. Note: M: DL2000 DNA marker, H1: *P. megaterium*, I2: *B. subtilis*, NC: negative control.

**Figure 2 microorganisms-12-00896-f002:**
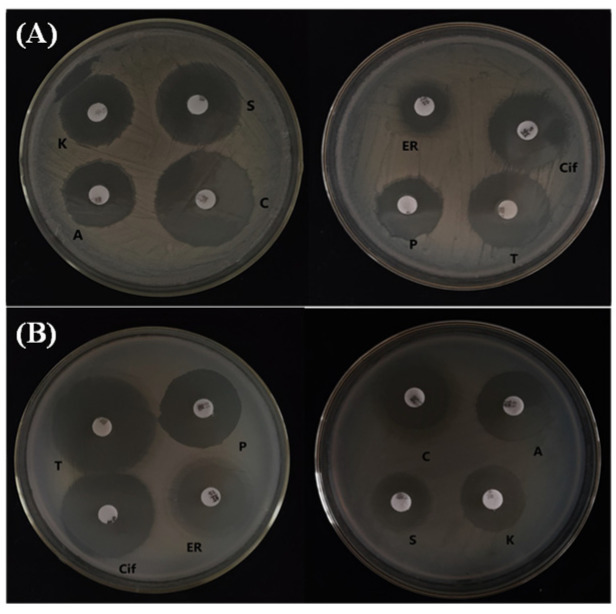
Antibiotic sensitivity test result of isolate (**A**) *P. megaterium* H1 and (**B**) *B. subtilis* I2. Note: Cif, *ciprofloxacin*; A, *ampicillin*; E, *erythromycin*; P, *penicillin*. C, *cefotaxim*; K, *kanamycin*; T, *tetracycline*; S, *streptomycin*.

**Figure 3 microorganisms-12-00896-f003:**
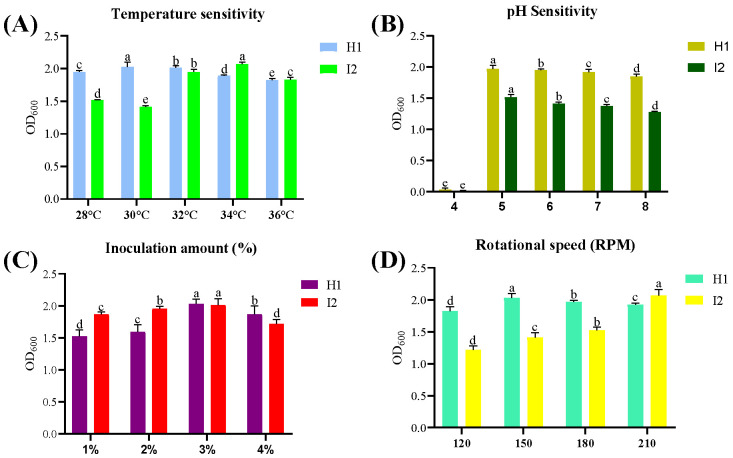
Effect of four different factors ((**A**) temperature, (**B**) pH, (**C**) inoculum size, (**D**) rotation speed) on the growth of antagonistic bacteria *B. subtilis* I2, *P. megaterium* H1 after 24 h of incubation. Error bars presented are the mean ± standard deviation of triplicates of three independent experiments. Different letters shows significant difference (*p =* 0.05).

**Figure 4 microorganisms-12-00896-f004:**
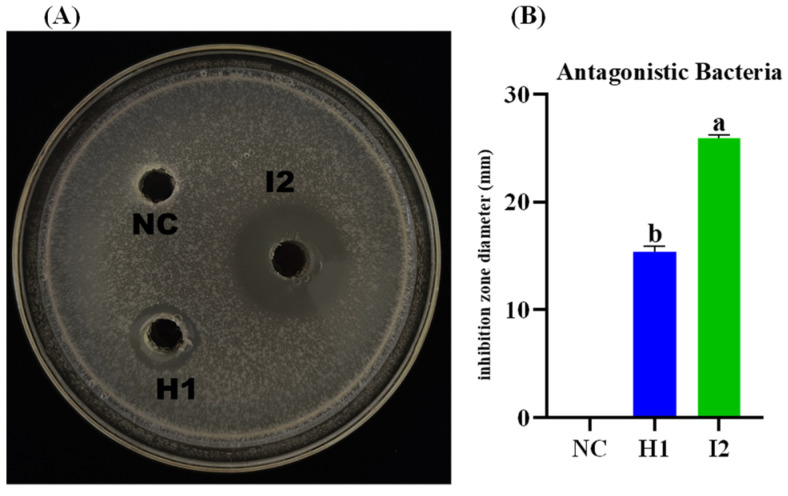
Screening and inhibition of *E. amylovora* by antagonistic bacteria. (**A**) The effect of antagonistic bacteria against *E. amylovora*, *B. subtilis* I2, *P. megaterium* H1, NC: Negative control (**B**) Diameter of antagonistic inhibition zone of antagonist strains. Different letters shows significant difference (*p =* 0.05).

**Figure 5 microorganisms-12-00896-f005:**
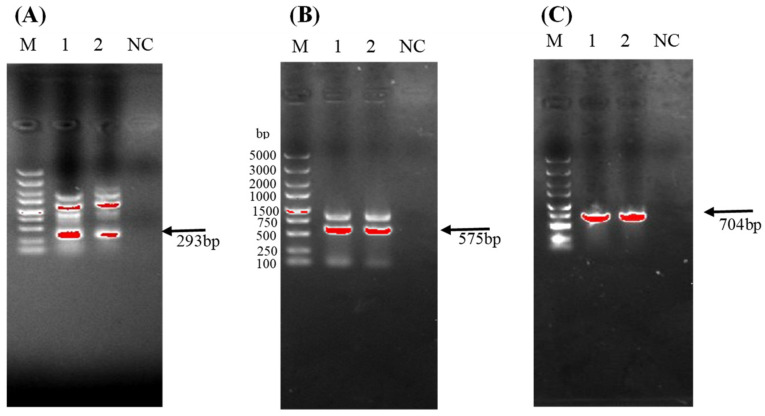
PCR gel electrophoresis of the *QK1*, *ItUc*, and *FenD* antibiotic synthesis genes. Note: (**A**) fenD, (**B**) *ituC*, (**C**) *QK1*. M: DL5000 DNA marker, 1: *P. megaterium* H1, 2: *B. subtilis* I2, NC: negative control.

**Figure 6 microorganisms-12-00896-f006:**
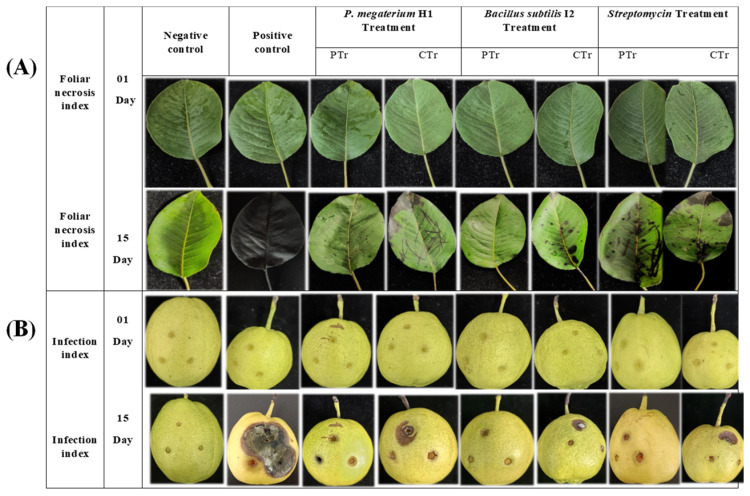
Protective and curative effects of *B. subtilis* I2 and *P. megaterium* H1 against *Erwinia amylovora* on detached pear leaves (**A**) and fruits (**B**) after 15 days post-inoculation (dpi). The leaves and fruits were collected from “Korla fragrant” pear.

**Figure 7 microorganisms-12-00896-f007:**
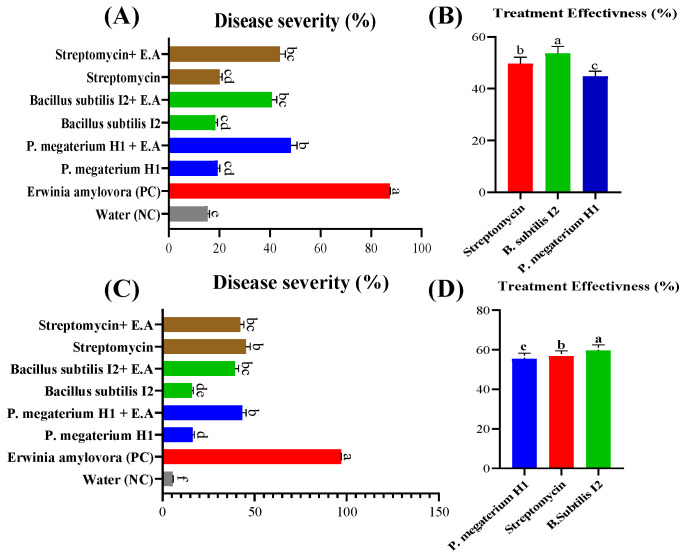
Disease severity and effectiveness of *B. subtilis* I2, *P. megaterium* H1, and the commonly used chemical compound (*streptomycin*) against *E. amylovora.* (**A**,**B**) fruits and (**C**,**D**) leaves. NC, negative control; PC, positive. Means with the same letter are not significantly different at *p* ≤ 0.05.

**Figure 8 microorganisms-12-00896-f008:**
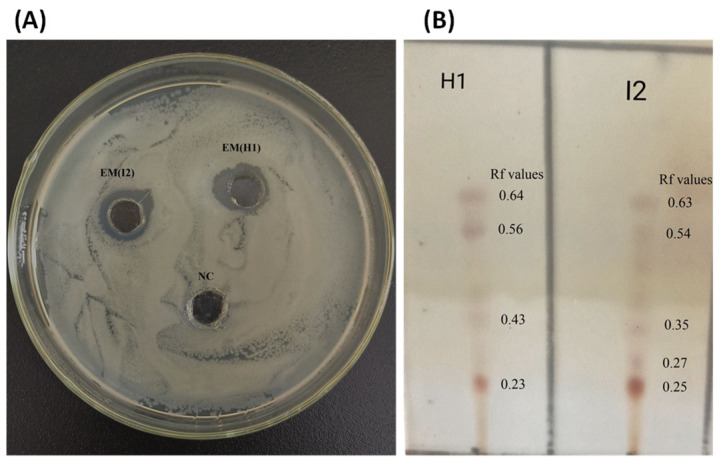
(**A**) The antibacterial activity analysis of organic extracts from *P. megaterium* strain H1 and *B. subtilis* strain I2 against *E. amylovora* on an NA medium. (**B**) Separation of active compounds by thin-layer chromatography (TLC). EM, extracted with methanol.

**Table 1 microorganisms-12-00896-t001:** List of RF values of various lipopeptides found in the TLC extract from H1 and I2.

S/No	Extract	RF Values (HI)	RF Values (I2)	Standard RF Values	Lipopeptides
1	Methanol	0.23	0.25	--	unknown
2	Methanol	--	0.35	0.35 ± 0.04	*Bacillomycin* D
3	Methanol	0.43	--	0.42 ± 0.04	*Fengycins*
4	Methanol	0.56	0.54	0.50 ± 0.04	*Iturins*
5	Methanol	0.64	0.63	0.62 ± 0.04	*Surfactins*

**Table 2 microorganisms-12-00896-t002:** Results of morphological and biochemical tests on isolates of *P. megaterium* H1 and *B. subtilis* I2.

S/n	Biochemical Test	Result for *B. subtilis* I2	Result for *P. megaterium* H1
1	Gram stain	+	+
2	Shape	Rod	Rod
3	Capsule	+	−
4	Motility	+	+
5	Colony color	White	White
6	Catalase test	+	+
6	Methyl Red test	−	−
8	Voges–Proskauer test	+	+
9	Simon citrate	+	+
10	Starch hydrolysis	+	+
11	Gelatin hydrolysis	+	+

Note: “+” and “−” represent positive and negative reactions, respectively.

## Data Availability

Data are contained within the article and Appendix A.

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
