# Peer review of "Bacterial Lipopeptides Are Effective against Pear Fire Blight"

_microorganisms, 2024, doi:10.3390/microorganisms12050896_

Round 1
Reviewer 1 Report
Comments and Suggestions for Authors
Overall, authors presented detailed manuscript, with interesting and relevant topic. However, text should be revised and corrected, as there is some grammar and punctuation mistakes left, which reduce the quality of the manusript and make it difficult to read.
Also, authors should unify MR and VP tests abreviations in methodology (Lines 139-140) and Results (Lines 299-300).
Author Response
Reviewer# 1:
Overall, authors presented detailed manuscript, with interesting and relevant topic. However, text should be revised and corrected, as there is some grammar and punctuation mistakes left, which reduce the quality of the manusript and make it difficult to read.
Re: Thank you for your positive comments. We have carefully considered the advice and revised our manuscript in detail. We hope that this manuscript is much improved now.
Also, authors should unify MR and VP tests abreviations in methodology (Lines 139-140) and Results (Lines 299-300).
Re: Thank you very much for comments. We modified the section accordingly and added abbreviations where was necessary. Line number 139,140,298 and 299.
Again, thank you for your positive and valuable comments. Certainly, it helps with improving the quality of this MS.
Reviewer 2 Report
Comments and Suggestions for Authors
Authors did extensive studies on isolation of bacteria antagonistic to E. amylovora that cause disease in pears. The research design and results are impressive, however I have some comments on the methodology:
-line 125 - what was the OD?
-chapter 2.3 - it would be beneficial to include antagonistic bacteria with documented activity (e.g., the once you mentioned in the introduction) against E. amylovora as positive control
-line 140 - this sentence is unnecessary here and better fitted in the last paragraph of introduction where you describe what was done in this study
line 180 - please include the name of the bacterium
line 182 - please indicate the concentration of bacteria (e.g., cell number or OD)
Minor editing issues:
- in Abstract please use italics for strain names
Author Response
Reviewer# 2:
Authors did extensive studies on isolation of bacteria antagonistic to E. amylovora that cause disease in pears. The research design and results are impressive, however I have some comments on the methodology
Re: Thank you for your positive comments. We have carefully considered the advice and revised our manuscript in detail. We hope that this manuscript is much improved now.
line 125 - what was the OD?
Re: Thank you very much for your comment. The OD value is OD600nm = 1 (approximately 1.0×108 - 1.0×109 CFU/mL) and added in the corresponding section. Line- 125.
chapter 2.3 - it would be beneficial to include antagonistic bacteria with documented activity (e.g., the once you mentioned in the introduction) against E. amylovora as positive control.
Re: Thank you for your constructive comment it has been added. Line 132-133
line 140 - this sentence is unnecessary here and better fitted in the last paragraph of introduction where you describe what was done in this study
Re: Thanks, we have modified the section and merged it into the introduction last paragraph. Line. 88-90
line 180 - please include the name of the bacterium.
Re: Thank you, the name has been added. Line- 179
line 182 - please indicate the concentration of bacteria (e.g., cell number or OD)
Re: Thanks, the relevant information has been added: OD600 = 1 (approximately 1.0×108 CFU/mL). Line- 182.
in Abstract please use italics for strain names
Re: Thanks, the strain names have been italicized. Line. 14,15,18 and 24.
Again, thank you for your positive and valuable comments. Certainly, it helps with improving the quality of this MS.
Round 2
Reviewer 2 Report
Comments and Suggestions for Authors
Thank you for providing revision. Again, I believe this study is thoroughly planned and conducted so I reccomend for publishing.